# Reproducibility in Machine Learning for Health

**Matthew B.A. McDermott**[*]
Massachusetts Institute of Technology
mmd@mit.edu

**Shirly Wang**[*]
University of Toronto
shirlywang@cs.toronto.edu

**Nikki Marinsek**
Evidation Health, Inc.
nmarinsek@evidation.com

**Rajesh Ranganath**
New York University
rajeshr@cims.nyu.edu

**Marzyeh Ghassemi**
University of Toronto
marzyeh@cs.toronto.edu

**Luca Foschini**
Evidation Health, Inc.
luca@evidation.com

## Abstract

Machine learning algorithms designed to characterize, monitor, and intervene on human health (ML4H) are expected to perform safely and reliably when operating at scale, potentially outside strict human supervision. This requirement warrants a stricter attention to issues of reproducibility than other fields of machine learning. In this work, we conduct a systematic evaluation of over 100 recently published ML4H research papers along several dimensions related to reproducibility. We find that the field of ML4H compares poorly to more established machine learning fields, particularly concerning data and code accessibility. Finally, drawing from success in other fields of science, we propose recommendations to data providers, academic publishers, and the ML4H research community in order to promote reproducible research moving forward.

## 1 Introduction

Science requires reproducibility, but many sub-fields of science have recently experienced a reproducibility crisis, eroding trust in processes and results and potentially influencing the rising rates of scientific retractions [1, 5, 44]. Reproducibility is also critical for machine learning research, whose goal is to develop algorithms to reliably solve complex tasks at scale, with limited or no human supervision. Failure of a machine learning system to consistently replicate an intended behavior in a context different from which that behavior was defined may result in dramatic, even fatal, consequences [27]. Ranking prominently among machine learning applications that may put human lives at stake are those related to Machine Learning for Health (ML4H). In a field where applications are meant to directly affect human health, findings should undergo heavy scrutiny along the validation pipeline from research findings to applications deployed in the wild. For example, in 2018, 12 AI tools using ML4H algorithms to inform medical diagnosis and treatment were cleared by Food and Drug Administration (FDA) and will be marketed to and potentially used by millions of Americans [31]. Verifying the reproducibility of the claims put forward by the device manufacturer should thus be a main priority of regulatory bodies [36], extending the need for reproducible ML4H results beyond the machine learning research community.

Unfortunately, several factors relating to the availability, quality, and consistency of clinical or biomedical data make reproducibility especially challenging in ML4H applications. In this work,

---

[*]Equal Contribution

we make several contributions. First, we present a taxonomy of reproducibility tailored to ML4H applications, and designed to capture reproducibility goals more broadly. Second, we use this taxonomy to define several metrics geared towards quantifying the particular challenges in reproducibility faced within ML4H, and conduct a comprehensive review of the published literature to support our claims and compare ML4H to machine learning more generally. Finally, we build on this analysis by exploring promising areas of further research for reproducibility in ML4H.

## 2  A Reproducibility Taxonomy

The common understanding of reproducibility in machine learning can be summed up as follows: A machine learning study is *reproducible* if readers can fully replicate the exact results reported in the paper. We will call this concept *technical replicability*, as it is centrally concerned with whether or not one can replicate the precise, technical results of a paper under identical conditions. Though intuitive, we argue technical replicability is actually only a small part of the goal of "reproducibility" more generally. This discrepancy has been noted historically in various ways [6, 20, 38] and is made apparent by the use of the term in the natural and social sciences, where attempted reproductions will often occur at different labs, using different equipment/staff, etc. We argue that in order for a study to be fully reproducible, it must meet three tiers of replicability:

**Technical Replicability**  Can results be replicated under *technically* identical conditions?

**Statistical Replicability**  Can results be replicated under *statistically* identical conditions?

**Conceptual Replicability**  Can results be replicated under *conceptually* identical conditions?

*Technical replicability* refers to the ability of a result to be fully technically replicated, yielding the precise results reported in the paper. For example, this entails aspects of reproducibility related to code and dataset release. *Statistical replicability* refers to the ability of a result to hold under re-sampled conditions that yield different technical configurations, but should not statistically affect the claimed result (e.g., a different set of random seeds, or train/test splits). Note this is related, but not identical, to the notion of *internal validity* [3] commonly used in social science research. Similarly, *conceptual replicability* is heavily related to *external validity* [37], as it describes the notion of how well the desired results reproduces under conditions that mach the conceptual description of the purported effect. Note that this is task-definition dependent; claiming a task has a greater conceptual horizon of generalizability makes it harder to satisfy this reproducibility requirement.

All three of these replicability criteria are central for full reproducibility: without technical replicability, one's result cannot be demonstrated. Without statistical replicability, one's result will not reproduce under increased sampling and the presence of real-world variance. And lastly, without conceptual replicability, one's result does not depend on the desired properties of the data, but instead depends on potentially unobserved aspects of the data generation mechanism that, critically, *will not reproduce when deployed in practice*.

Under each of these lenses, ML4H differs from general machine learning domains in critical ways and presents unique challenges.

## 3  Core Reproducibility Challenges in ML4H

In this section, we illustrate both through qualitative arguments and a quantitative literature review that ML4H lags behind other subfields of machine learning on various reproducibility metrics. Our literature review procedure entailed manually extracting and annotating over 300 papers from various venues, spanning ML4H, Natural Language Processing (NLP), Computer Vision (CV), and general machine learning (general ML).[2] The full procedure used for this statistical review is detailed in the Appendix (Section 5), and final quantitative results on several key metrics can be seen in Figure 1, though they are also detailed in the text where appropriate.

---

[2]NLP and CV were chosen to represent broad fields with a significant applied focus, much like ML4H. General ML was chosen to have an unbiased comparison to the field more broadly.

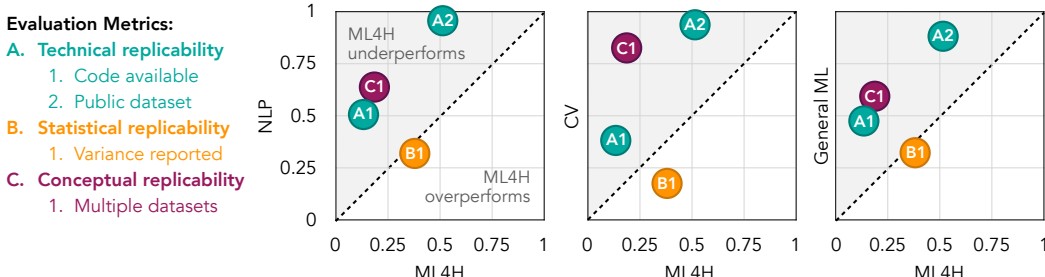

Figure 1: Fraction of papers satisfying certain conditions by ML field. See the Appendix (Section 5) for detailed descriptions of the underlying data collection procedure. Note that ML4H consistently lags other subfields of machine learning on all measures of reproducibility save inclusion of proper statistical variance.

## 3.1   Technical Replicability

ML4H faces several key challenges with regards to technical replicability. Firstly, health data is privacy sensitive, making it difficult to release openly without either incurring risks of re-identification, or severely diminishing utility by applying aggressive de-identification techniques [7]. As a result, few public datasets are available, and those that are available are used extremely frequently, leading to a risk of dataset-specific over-fitting. To this point, approximately only 51% of the ML4H papers we examined used public datasets, as compared to over 90% of both CV and NLP papers, and approximately 77% of general ML papers.

ML4H scores even more poorly when it comes to code release, preprocessing specification, and cohort description; only approximately 13% of the papers we analyzed released their code publicly, compared to approximately 37% in CV and 50% in NLP. Specific prior works have also already examined the prevalence of code release and dataset cohort sub-selection within ML4H, finding that even when restricting focus to public datasets, studies often do not release code or include sufficiently informative text to enable a full technical reproduction [16, 23]. Note that code release is itself not necessarily sufficient for full technical replicability; even when code is released, it may not run correctly, it may exclude critical details, or it may fail to generate the results reported in the paper.

## 3.2   Statistical Replicability

To assess the state of statistical replicability in ML4H, we quantified how often papers describe the variance around their results (e.g., by listing both the mean and the standard deviation of a performance metric over several random splits). Interestingly, while ML4H's rate of this is still relatively low (approximately 38%), it is higher than that of CV, NLP, or the general domain (17%, 33%, and 32%, respectively).

Though this is an encouraging sign, we feel there is still significant room for improvement. Even in other fields of machine learning, with arguably less complex data types, repeated studies have shown that published models fail to statistically reproduce when appropriate statistical procedures are implemented and fair hyperparameter search/preprocessing methods are used: in [41], researchers find that published models evaluated using the public ImageNet test set show consistent performance drops when trained/tested on other random splits within ImageNet, and in [30, 29], researchers find that published performance deltas between various model architectures for language models and generative adversarial networks, respectively, fail to persist under more robust hyperparameter search and statistical comparison techniques. In [17], researchers examine practices in deep reinforcement learning which can limit or enhance studies' reproducibility. Due to our inability to technically replicate many ML4H works given dataset and code release issues, we cannot assess the extent of these issues to the same degree in ML4H. However, it seems doubtless that similar problems affect our field, perhaps to an even greater degree given that our datasets are frequently relatively small, high dimensional, sparse/irregularly sampled, and suffer from high rates of noise.

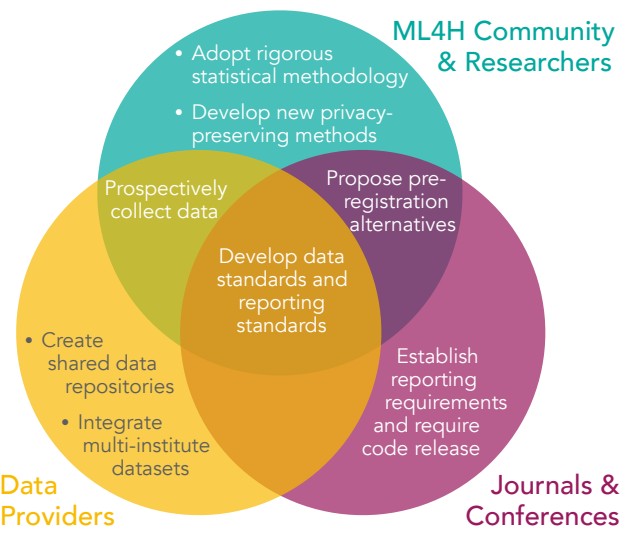

Figure 2: Summary of recommendations for different stakeholders.

### 3.3 Conceptual Replicability

The critical issue in ML4H preventing conceptual replicability is the lack of multi-institution datasets in healthcare and the lack of usage of those that do exist. Whereas approximately 83% of CV studies and approximately 66% of NLP studies used multiple datasets to establish their results, only approximately 19% of ML4H studies did. Using only a single dataset is potentially dangerous, as it is known that developing ML4H models that generalize over changing care practices or data formats is challenging. In [13], researchers demonstrate the dangers of training a model on raw data from one institution and transferring to another (using a simulated institutional divide within the MIMIC dataset), and in [34], researchers establish that without using manually engineered representations, ML4H models exhibit significant degradation in performance over time as care patterns evolve. These results are not surprising; health data is rife with hidden confounders, differs significantly between data collection and deployment environments, drifts over time, and further differs in structure and concept between different healthcare institutions [4].

## 4 Opportunities for Improvement

In this section, we present several practical suggestions for enhancing reproducibiltiy for ML4H from the perspective of three main ML4H stakeholders: the ML4H research community, data providers, and related journals and conferences. Figure 2 summarizes these suggestions.

### 4.1 Create Shared Research Resources

Data providers such as hospitals, clinical research centers, and government agencies produce vast amounts of valuable data. Unfortunately, as suggested by our literature review, few datasets are available for a wide range of researchers to explore. We call for more instances of large data trusts where medical institutions can anonymously pool data for researchers to use and create algorithms. Several prominent examples already exist for the field to draw on in creating such repositories, including MIMIC [24], the U.K and Japan Biobanks [32, 43], eICU [39], the Temple University Hospital EEG Corpus [15], and Physionet [12]. This recommendation is especially compelling as corporate entities increasingly invest in the ML4H space, particularly with regards to data [40, 49]. Anchoring our research progress to large scale datasets in non-public or non-academic hands poses a dangerous precedent, especially in light of recent patent filings [2, 14].

### 4.2 Integrate Multi-Institute Datasets

Multi-institute datasets enable studies to assess their ability to translate to new contexts, a critically understudied facet of ML4H research. Recent strides have been taken in this domain with the release of the eICU dataset [39], one of ML4H's first large scale, multi-institution EHR datasets, and researchers are already analyzing how to form generalizable models using this resource [21]. Lastly, Observational Health Data Sciences and Informatics (OHDSI) [19] provides a mechanism to run observational health studies across multiple institutions and countries. We encourage more collaborative efforts in this area from data providers to improve conceptual replicability.

### 4.3 Prospectively Collect Data

Data collected as a by-product of care, then later released for research purposes (e.g., MIMIC [22]), presents serious privacy risks and contains many confounding variables. The landscape of these privacy risks and the nature of the confounding variables change (though they do not necessarily lessen) if data is instead prospectively collected directly from new, consented participants. These types of data collection regimes are logistically challenging, but are possible. The NIH's All of Us Research Program [33], Evidation's DiSCover Project [9], and Google's Project Baseline [46] are examples of these approaches.

### 4.4 Adopt Rigorous Statistical Methods

ML4H researchers should be more rigorous in the development, refinement, and dissemination of statistical best practices (e.g., the proper procedures for model comparisons, etc.). Holding ourselves to high standards of statistical rigor, potentially including periodic statistical audits of our own statistical replicability, will help ensure we mitigate the problems other fields have found with "over-fitting via publication," e.g. [41]. Several commonly-used challenges already exist in ML4H with fixed train/test sets (e.g., [11, 42]) which could be used for these kind of post-hoc replication studies.

### 4.5 Develop New Privacy-Preserving Analysis Techniques

Technological solutions can also be employed to help mitigate privacy concerns. ML4H researchers can explore noised, fully- or partially- simulated, or encrypted datasets. In cases where data cannot be released, techniques to train distributed models without sharing data have been proposed [45]. Synthetic data can be an excellent tool to help enable researchers to still meaningfully release their code with full end-to-end realizations of their pipeline, as is done in, e.g., [50]. Technology for producing synthetic patient-level data also already exists [47].

### 4.6 Propose Pre-Registration Alternatives

In the biomedical sciences, "observational studies," under which definition nearly all ML4H research falls, undergo intense scrutiny to ensure they are not susceptible to statistical artifacts—in particular, increasingly these studies are required to be *pre-registered*,[3] a move promoted by scientists and major journals alike [25, 28, 48]. Such prospective checks make unintentional statistical fraud more difficult, but are utterly absent in ML4H. A verbatim application of the pre-registration practice in use in the epidemiology community is likely an impractical solution due to the intrinsic exploratory nature of model development, but other techniques, such as systematic re-release of new data and/or rotation of official train/test splits have been found to help reduce the presence of statistical errors in other fields. ML4H researchers and academic publishers should engage in serious conversation on what the best vehicle for these checks and balances are.

### 4.7 Establish Reporting Requirements and Require Code Release

If conferences and journals required greater rates of data/code release, or the use of additional data/code availability statements at publication, this would instill a significant pressure on the field

---

[3] Meaning they are required to report their goal and planned preprocessing/modeling scheme before they run any experiments in order to avoid intentional or unintentional statistical fraud. This idea is not without its detractors [26, 8].

to address some of the foundational barriers to reproducibility. Conferences and journals have the additional ability to insist on high standards for the reporting of statistical variance around results, hyperparameter search procedures, and evaluation mechanisms, each of which would help ensure that we maintain high standards of technical and statistical replicability.

### 4.8 Develop Data Standards and Reporting Standards

Collaborative efforts in developing data standards and reporting standards is another avenue for improving reproducibility. Healthcare analytics organizations have developed data standards like the OMOP standard [35] or the FHIR standard [18], but they are not commonly adopted in ML4H research. Increased use of standards would make it easier to technically and conceptually replicate ML4H studies. Similarly, when ML4H datasets are created, we need greater descriptions of their contents, potential confounders and biases, missing data prevalence and distribution, and how they were created. Increasing the use of "specs" or "datasheets" describing datasets would help allay these concerns—efforts in the broader machine learning community are also approaching this goal [10] and we should vigorously adopt these practices in ML4H.

## 5 Conclusion

In this work, we have framed the question of reproducibility in ML4H around three foundational lenses: technical, statistical, and conceptual replicability. In each of these areas, we argue both qualitatively and quantitatively, through a manual, extensive review of the literature, that ML4H performs worse than other machine learning fields in several reproducibility metrics we have identified. While keeping in mind the intrinsic challenges of data acquisition and use that plague the field, we highlight several areas of opportunities for the future, focused around improving access to data, expanding our trajectory of statistical rigor, and increasing the use of multi-source data to better enable conceptual reproducibility.

## Acknowledgments

This paper benefited substantially from the help of many people. Most notably, Bret Nestor, Amy Lu, Denny Wu, Elena Sergevea, and Di Jin all helped annotate papers for our analysis.

Additionally, this work was funded in part by National Institutes of Health: National Institutes of Mental Health grant P50-MH106933, as well as by University of Toronto CIFAR Chair support.

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

## Appendix: Statistical Review Procedures

**Selection Criteria** Papers were selected at random, to ensure an unbiased sample, from various venues associated with different domains (though papers were tagged with their content-driven domain at annotation time). A full list of venues, along with the number of papers analyzed from each, is presented in Table 1. All publicly-accessible papers within the ML4H venues were used, rather than a random sub-sample, given the venues' limited sizes. Except for papers from MLHC2017 and ICCV, all papers were published in 2018. We conducted a similar review for 2018 papers only and the results were consistent.

**Annotation Procedure** Each paper was reviewed by only one annotator, and in total 8 annotators were used, 3 of which are authors on this work. Annotation tasks were designed to be quick, to enable us to profile a large number of papers at a coarse level rather than a small number of papers in depth. As such, annotators were only asked the following questions:

1. Was code released?

2. What datasets were used?

3. Are these datasets publicly available (modulo data use agreements)?

4. Do the authors report any notion of variance around their results or assess their comparisons to baselines in a statistically robust fashion (e.g., via hypothesis testing)?

| Domain | Conference | # Papers Examined |
|--------|-----------|-------------------|
| ML4H | MLHC 2017 | 25* |
|  | MLHC 2018 | 31* |
|  | ML4H Poster-Accepts 2018 | 57* |
|  | KDD Health Day 2018 | 21* |
| NLP | ACL | 30 |
|  | EMNLP | 30 |
|  | NAACL | 30 |
| CV | CVPR | 21 |
|  | ICCV | 21 |
|  | ECCV | 21 |
| ML | NeurIPS | 14 |
|  | ICML | 14 |
|  | ICLR | 14 |
|  | KDD | 14 |
|  | AAAI | 14 |

Table 1: Sources and coverage statistics for our manual literature review. * indicates all publicly-accessible papers published were used.

**Potential Biases** This selection and annotation procedure allowed us to analyze a large number of papers, but has several possible biases. In particular, our annotation questions were all of these were designed to be determinable via quick, scanning techniques and as a result this task took on average between 45 seconds and 3 minutes per paper. In such a limited time, some losses are unavoidable. We recognize several sources of possible bias worth mentioning.

Firstly, some papers may, for example, release datasets or code products external to the paper and not mention it in the actual text. We will omit these associated products. If such effects induce a notable bias in our results, however, we must question why as a field we are comfortable releasing our code/data without any mention in the associated paper.

Secondly, not all papers intended to be analyzed were publicly accessible. Similarly, the versions of papers we analyzed could have been different from the version presented at the actual conference venue, or there could exist updated versions of papers we analyzed in different repositories. Our analysis technique will miss these effects.

Thirdly, some papers naturally fit into multiple categories (e.g., a work focused on medical named entity recognition would be both a ML4H work and an NLP work). In the interest of ensuring our comparison classes were as pure as possible, we omitted all clearly multi-domain works, but allowed works that centered primarily in a single domain to remain.

Lastly, different fields present different kinds of works, and not all works fit into our framework. Largely theoretical works, for example, often have no real datasets or public experiments. Similarly, presenting variance is a different question for works focused principally around computational efficiency rather than predictive accuracy. We handled these issues by attempting to answer these questions as best we could, and flagging any papers that overtly did not fit our scheme and excluding them from our analyses.

**Data Release** We release all data as a result of these analyses publicly, accessible at the time of publication.

