# OpenReview forum: "Reproducibility in Machine Learning for Health"
_ICLR.cc/2019/Workshop/RML — RML 2019_

### Official Review · AnonReviewer1 · 2019-04-01
**Interesting work on ML4H Reproducibility**

**Rating:** 5
**Confidence:** 3

**Review:**

Summary: This paper conducted an excellent quantitative and qualitative review of the state of the reproducibility for ML healthcare applications.  I learned a great deal from reading it!

Notes:
  -Reproducibility is especially important in health due to safety concerns.
  -Review of 100 ML4H research papers relating to reproducibility
  -ML4H has more issues with data and code access.
  -Proposes recommendations to make research more reproducible.
  -In 2018, 12 healthcare tools using ML got FDA clearance.  (cool!)
  -Quantitative and qualitative review showing ML4H has worse code availability data availability and dataset variety than other ML subfields.
  -The choice of evaluation metrics is reasonable but a bit limited.
  -Privacy issues make it difficult to release health data publicly.
  -ML4H papers are more likely to report mean/stdv than other fields in ML.
  -Only 19% of ML4H studies used multiple datasets.
  -The issue of preregistration in ML is interesting.

Comments / questions:
  -Does failure to reproduce basic research papers in ML4H really lead to problems for production health systems?  Presumably there are many steps of validation beyond the basic research papers.
  -I really like the reproducibility taxonomy: technical, statistical, and conceptual reproducibility.  Technical reproducibility refers to getting the exact results, so includes things like implementation.  Statistical reproducibility is equivalence but only up to the statistical properties being the same (so the results follow the same distribution).  Conceptual reproducibility means that the idea works as long as the concept is preserved.
  -Figure 1 is quite nice

---

### Decision · Program_Chairs · 2019-04-05
**Acceptance Decision**

Accept